# Dynamic Shifts in Antibiotic Residues and Gut Microbiome Following Tilmicosin Administration to Silkie Chickens

**DOI:** 10.3390/ani14233428

**Published:** 2024-11-27

**Authors:** Qiying Liang, Chunlin Xie, Haile Berihulay Gebreselase, Yushan Yuan, Jingyi He, Lu Xie, Chenglong Luo, Jian Ji

**Affiliations:** 1State Key Laboratory of Swine and Poultry Breeding Industry, Guangdong Key Laboratory of Animal Breeding and Nutrition, Institute of Animal Science, Guangdong Academy of Agricultural Sciences, Guangzhou 510640, China; lennonliang@yeah.net (Q.L.); cylinqueen@163.com (C.X.); haile.berihulay@gmail.com (H.B.G.); yuanyushan2017@163.com (Y.Y.); hjypuppy@163.com (J.H.); xl115816@163.com (L.X.);; 2Department of Animal Science, South China Agricultural University, Guangzhou 510642, China

**Keywords:** tilmicosin, gut microbiota, blood biochemistry, Silkie chicken

## Abstract

Antibiotics are commonly used in animal production to treat or prevent bacterial infections. Tilmicosin is one of the antibiotics widely used in livestock and poultry. However, the use of antibiotics raises concerns about the development of antibiotic-resistant pathogens due to the strong selection pressure. One common approach to mitigate the antibiotic residue risk is implementing withdrawal periods. However, there is a trade-off between animal growth time and profit. Therefore, alternative methods for reducing antibiotic residue in animal products have garnered significant interest. In this study, we systematically analyzed the changes in the concentration of drug residues in different tissues of Silkie chickens for 40 days post tilmicosin treatment. Furthermore, we explored their effects on blood biochemical parameters for 140 days post-treatment. We conducted 16S rRNA sequencing to uncover the long-lasting effects of tilmicosin treatment on microbial diversity and composition from 1 to 120 days. Finally, correlation analysis between microbiota and concentrations of antibiotic residues in different tissues indicated that *Mucispirillum_schaedleri*, *Parabbacteroide_distasonis*, and *Faecalibacterium_prausnitzii* were strongly associated with tilmicosin residue.

## 1. Introduction

Tilmicosin (TIM), a semi-synthetic macrolide antibiotic, is widely used in livestock and poultry to treat bacterial infections [1]. The unique structure of TIM blocks bacterial ribosome activity and inhibits their protein synthesis. As a result, TIM demonstrates strong antimicrobial efficacy in vitro against Gram-positive and select Gram-negative bacteria [2]. Its approval for treating respiratory diseases in chickens caused by *Mycoplasma gallisepticum* and *Mycoplasma synoviae* highlights its practical significance [3,4]. However, the widespread use of antibiotics in food animal production leads to the accumulation of drug residues in animal products, posing a significant threat to both food safety and public health [5]. Moreover, it creates conducive environments for the proliferation of antibiotic-resistant bacteria [6,7,8].

The gut microbiota is crucial for the development of the immune system, maintaining intestinal homeostasis [9], and actively participating in nutrient metabolism [10], ultimately influencing the overall performance of livestock and poultry [11]. Moreover, it is acknowledged for its role in mediating various biotransformation activities of xenobiotic compounds, including therapeutic drugs [12]. Previous research has revealed that a significant proportion of drugs, specifically 66% out of 271, can be metabolized by at least one type of bacteria in humans [13]. While TIM has been shown to effectively reduce the incidence and severity of airsacculitis lesions caused by *Mycoplasma gallisepticum* [14], the complex interaction between TIM and the gut microbiome remains unclear.

Silkie chicken, renowned for its flavor and nutritional value, is an important indigenous breed and valuable germplasm resource [15]. However, the use of antibiotics in Silkie chicken production presents significant concerns. This study aims to investigate the relationship between TIM residues and gut microbiota in the Silkie chicken model (Figure 1). Initially, we examined the pattern of TIM residue removal in various tissues and assessed blood biochemical parameters after TIM treatment to understand its effects on chicken physiological metabolism. Subsequently, we conducted a comprehensive comparison of the characteristics of gut microbial composition and diversity using 16S rRNA sequencing. Finally, through correlation analysis, we successfully identified three microbes, *Mucispirillum_schaedleri*, *Parabbacteroide_distasonis*, and *Faecalibacterium_prausnitzii*, which were strongly associated with TIM residue removal. Therefore, our research highlights the important role of gut microbiota in influencing antibiotic metabolism, providing a potential strategy for reducing drug residues in animal food and ensuring food safety.

## 2. Materials and Methods

### 2.1. Silkie Chickens and Study Design

In this study, a total of 250 female Silkie chickens were purchased from Xufeng Agricultural Ltd., Guangzhou, China. All chickens were given feed ad libitum. The 1-day-old to 30-day-old chicks were offered feed 531 N (ZhengDa Co., Ltd., Yangjiang, China) with the following nutrition composition: crude protein ≥ 20.0%, crude fiber ≤ 6.0%, crude ash ≤ 8.0%, calcium in the range of 0.60~1.80%, phosphate ≥ 0.40%, sodium chloride in the range of 0.20~0.80%, and lysine ≥ 1.00%. The 31-day-old to 120-day-old chickens were offered 532N (ZhengDa Co., Ltd., Yangjiang, China) with the following nutrition composition: crude protein ≥ 16.0%, crude fiber ≤ 7.0%, crude ash ≤ 8.0%, calcium in the range of 0.60~1.30%, phosphate ≥ 0.40%, sodium chloride in the range of 0.20~0.80%, and lysine ≥ 0.80%. Crude protein, crude fiber, crude ash, and minerals were determined in accordance with AOAC [16]. The analytical procedures employed specific equipment, including precision balance (BSA2245-CW, Sartorius AG, Göttingen, Germany), drying chamber (Shanghai YiHeng Scientific Instrument Ltd., Shanghai, China), muffle furnaces (WMF-05, Wiggens Co., Ltd., Wuppertal, Germany), Kjeldahl nitrogen analyzer (K12, Sonnen Automated Analysis Instrument Co., Ltd., Shanghai, China), and atomic absorption spectrophotometry (ZCA-1000, ZhongHeCeTong Ltd., Beijing, China). Lysine was determined by amino acid analyzer L-8900 (Hitachi High-Tech Co., Tokyo, Japan). To examine the effect of TIM (Wens Dahuanong Biotechnology Co., Ltd., Yunfu, China) residue on chickens, we randomly divided the 30-day-old chickens into two groups: the control group and the treatment group. TIM, included in the drinking water at a concentration of 0.75 g/L, was administrated to the treatment group at the beginning of the experiment from day 1 to day 4. After the treatment, 10 chickens from the treatment group and 6 chickens from the control group were randomly sacrificed at fourteen different time points (4 h, 1, 3, 5, 7, 9, 12, 20, 30, 40, 50, 60, 75, and 120 days). Individual blood samples were collected using sodium heparinized tubes. Samples of the left thorax skin tissues, left chest muscle, liver, and kidney were collected using sterile instruments and stored in germ-free centrifuge tubes for drug analysis. Moreover, fecal samples from the mid-section of the cecum were collected using sterile tubes and stored in liquid nitrogen.

### 2.2. Determination of TIM Concentration

To calculate the concentration of TIM, each tissue sample (1 g) was mixed with 10 mL of Acetonitrile in a 50 mL centrifuge tube and vortexed for 5 min. After vortexing, the tubes were centrifuged at 8000 rpm for 5 min. The clear upper portion of the centrifuged liquid was extracted into a 10 mL centrifuge tube and subjected to a 50 °C water bath until it evaporated to 1 mL using nitrogen gas. The remaining 1 mL of liquid was mixed with an additional 1 mL of 50% methanol. Subsequently, 3 mL of hexyl hydride was added to separate the lipid. Then, 1 mL of the sublayer was extracted and centrifuged at 10,000 rpm for 2 min. The centrifuged sublayer was filtered using 0.22 um nylon filters.

The Shimadzu LC-MS 8050 system (Kyoto, Japan), consisting of the CBM 20A controller, LC 30AD pump, DGU 20A degasser, SPD M30A detector, SIL 30AC autosampler, CTO 30A column oven, and LCMS 8050 mass spectrometer, was utilized to determine the concentration of TIM in the extracted liquid. For chromatographic separation, a Kinetex C18 column (100 × 2.1 mm i.d., 2.6 μm; Phenomenex, Torrance, CA, USA) at a temperature of 40 °C was employed. An external calibration curve was created by injecting 5 μL of standardized TIM at various concentrations (10 ng, 20 ng, 50 ng, 100 ng, 200 ng, 500 ng, and 1000 ng). The mobile phase, consisting of methanol (8:2) and 0.1% formic acid, was operated at a flow rate of 0.3 mL/min. The mass spectra were recorded under the infusion mode using a full scan in positive ion mode. The target compound, TIM, was quantified at *m/z* 332.1 and *m/z* 340.1. The experiment and data analysis were conducted using the LabSolution software program (https://www.shimadzu.com/an/products/software-informatics/software-option/labsolutions-cs/index.html, accessed on 24 September 2024).

### 2.3. Biochemical Analysis

For the biochemical analysis, the plasma samples were centrifuged for 10 min at 4000 rpm, and the resulting supernatant was then stored at 4 °C. The Chemray 800 Biochemistry autoanalyzer was used in conjunction with various kits to determine the concentration of alkaline phosphates, bilirubin, urea nitrogen, uric acid (kit from Rayto Life and Analytical Science Co., Ltd., Shenzhen, China), bile acids (Changchun Huili Biotech, Changchun, China), and glutathione peroxidase (Nanjing Jiancheng Bioengineering Institute, Nanjing, China).

### 2.4. Microbial Genomic DNA Extraction

The Mag Pure Stool DNA KF kit B (Magen, Guangzhou, China) was used to extract bacterial 16S rRNA from the cecal digesta. The extraction procedures were carried out following the manufacturer’s manual. The quantity and quality of the extracted DNA were assessed using the Qubit fluorometer and the Qubit dsDNA BR Assay kit (Invitrogen, Carlsbad, CA, USA). Following extraction and quality assessment, DNA samples were stored at −20 °C until sequencing.

### 2.5. Microbial 16S rRNA Gene Sequencing

BGI-seq500 primers 515F (5′-GTGCCAGCMGCCGCGGTAA-3′) and 806R (5′-GGACTACHVGGGTWTCTAAT-3′) were used to create the binding sites for sequencing. The PCR conditions were as follows: 5 min of denaturation at 94 °C, followed by 10 cycles of 30 s at 94 °C, 30 s of annealing at 60 °C, 30 s of elongation at 72 °C, and a final extension at 72 °C for 10 min. Following amplification, the product was extracted with a 2% agarose gel, purified with a QIAquick Gel Extraction kit (Qiagen, Hilden, Germany), and quantified with the Qubit dsDNA HS Assay Kit (ThermoFisher, Waltham, MA, USA). The purified amplicons were then sequenced using the MGISEQ-2000 platform, with the pooled samples being in equimolar concentrations.

### 2.6. Sequencing Data Analysis

Raw data obtained from sequencing were processed to remove adapter sequences and regions of low quality. FLASH (Fast Length Adjustment of Short Reads, v1.2.11) was used to merge the paired-end sequence reads. The sequences were then clustered into operational taxonomic units (OTUs) at a 97% identity threshold using USEARCH (v7.0.1090) [15]. The Ribosomal Database Project (RDP) classifier and scripts in QIIME (v1.8.0) software were employed to assign the OTUs to specific taxonomic groups [16]. The Venn diagram and Upset plot were generated using TBtools (version 1.068). α-diversity metrics (observed OTU richness, Chao1 richness, and the Shannon and Simpson diversity indices) were calculated from the rarefied libraries with Mothur (v1.31.2), while rarefaction curves were created with R (v3.4.1) software. β-diversity analysis, involving unweighted UniFrac and weighted UniFrac, was conducted with QIIME (v1.8.0) software. Principal component analysis (PCA) was performed using the “vegan” package in R 3.4.1 (The R Foundation for Statistical Computing, Vienna, Austria).

### 2.7. Statistical Analysis

Prism v.8.0 (GraphPad Software Inc., La Jolla, CA, USA) was used for statistical analyses. The difference in blood biochemical indicators between the treatment and control groups was examined by a Student’s *t*-test. A *p*-value < 0.05 was considered significant. The α-diversity, β-diversity, and taxa abundances at the phylum and generic levels between the two groups were assessed using the Wilcoxon test and visualized with box-and-whisker plots. Differences in microbial composition between the groups were analyzed using permutational multivariate analysis of variance (PERMANOVA) with the “adonis” function in the R vegan package. *p*-values were corrected for multiple sampling using the Benjamini–Hochberg false discovery rate procedure with the p.adjust function in R.

## 3. Results

### 3.1. Patterns of TIM Residue Elimination Across Different Tissues

The concentrations of TIM in the chest muscle, skin, liver, and kidney exhibited variations at different time points (Figure 2). The level of TIM residue in the chest muscle peaked on day 1, then gradually dropped until it reached almost zero on day 30. The skin tissue had its highest TIM residue value on day 1 (543.837 µg/kg) and was gradually metabolized thereafter. TIM concentrations in the liver and kidney peaked at over 2000 ng/kg. Like the chest and skin tissues, the liver’s TIM residue reached its highest point on day 1 (2038.04 µg/kg). However, in the kidney, the TIM residue reached its maximum on day 0.16 (2286.64 µg/kg). In these four tissues, TIM quickly reached its peak concentration on day 1 or day 0.16 but took a long time to be completely metabolized.

### 3.2. Effect of TIM Treatment on Blood Biochemical Parameters

After the TIM treatment, we further assessed blood biochemical indices to investigate the impact of TIM residue on different organs. Alkaline phosphatase (ALP), direct bilirubin (DBIL), total bilirubin (TBIL), and total bile acid (TBA) were measured to assess hepatic function, while blood urea nitrogen (BUN) and uric acid (UA) were evaluated to indicate renal function. Additionally, glutathione peroxidase (GSH-PX) was measured to determine serum antioxidant capacity. Our result showed that, in the TIM treatment group, ALP significantly decreased on day 3 and day 7 (Figure 3A). Biomarkers DBIL and TBIL both increased on day 3 and then decreased on day 5 in the treatment group (Figure 3B,C). TBA, the indicator of chronic hepatic disease, significantly increased on day 0.16 and day 9 (Figure 3D). Renal function biomarkers BUN and UA showed significant reductions in diverse patterns. BUN decreased in the mid- and late-term, day 9, day 12, and day 60 (Figure 3E). UA decreased in the early and mid-period, day 3 and day 12 (Figure 3F). Moreover, the serum antioxidant capacity indicator GSH-Px increased on day 3 and then decreased on day 7 (Figure 3G), which has a similar pattern with DBIL and TBIL.

### 3.3. The Affect of TIM on Microbiota Structure

To explore the correlation between TIM and gut microbes, we performed 16S rRNA sequencing to analyze the diversity and composition of the cecal microbiome at various time points following TIM treatment. The number of OTUs gradually increased with age in both the treatment and control groups, with no significant difference observed between the two groups (Figure 4A). The Chao1 index, which reflects microbiota richness, showed significant differences on day 9, day 30, and day 50. This suggests that the treatment group had a less diverse microbiome in the early period (day 9), but increased diversity in the mid-period (day 30), followed by a decrease (day 50) (Figure 4B). However, the Shannon index, another indicator of richness, indicated an increase in microbiome diversity on day 5 (Figure 4C). Principal coordinates analysis (PCoA) demonstrated significantly different clustering of gut microbiota with aging (Figure 4D). Using the weighted Unifrac distance, we observed significant differences between the TIM group and the control group on day 3, day 20, and day 40 (Figure 4E).

### 3.4. Shifts in Microbial Composition After TIM Treatment

Shifts in microbial composition at the phylum level were investigated following treatment with TIM. On average, the dominant phyla that made up the cecum microbial ecology were Bacteroidetes (40.83%), Firmicutes (31.35%), Euryarchaeota (17.37%), and Proteobacteria (5.28%) (Appendix A). Comparative analysis with the control group revealed an increased abundance of Actinobacteria (days 9, 12, 30), Deferribacteres (days 3, 5, 7, 9, 12, 20), Lentisphaerae (day 30), Proteobacteria (days 20, 120), Synergistetes (days 12, 20), and Verrucomicrobia (day 30) following TIM treatment (Figure 5). However, Bacteroidetes increased on days 7, 9, and 12 but decreased on day 60. Additionally, Cyanobacteria (day 7) and Firmicutes (days 9 and 12) exhibited decreased abundance in the TIM treatment group (Figure 5A–J).

At the genus level, *Methanobrevibacter* (17.21%), *Bacteroides* (16.00%), *Phascolarctobacterium* (4.36%), *Faecalibacterium* (2.33%), and *Oscillospira* (2.03%) were the dominant bacteria (Appendix A). Interestingly, the changes in *Bacteroides* (day 5), *Methanobrevibacter* (day 3), and *Oscillospira* (day 1) were significantly correlated with TIM (Figure 6A). Moreover, the relative abundance of *Butyricicoccus* (day 5–40), *Odoribacter* (day 1–20), *Mucispirillum* (day 5–20), and YRC22 (day 5–30) showed prolonged alterations (over 15 days) following TIM treatment. These results suggest that TIM treatment has a sustained impact on the gut microbiome.

Conversely, TIM treatment significantly reduced the population of certain microbial variants (Figure 6B). *Bacteroides*, *Methanobrevibacter,* and *Prevotella* were notably the top three microbes with the lowest relative abundance after TIM treatment. Bacteroides experienced a decline from day 1, reaching their lowest point on day 5, with a relative abundance ratio of 1:18.5 compared to the control group. Both *Methanobrevibacter* (day 3) and *Prevotella* (day 40) saw their numbers diminish to one-third of their original levels following TIM treatment. In addition to these microbes, TIM treatment consistently suppressed the populations of *Butyricicoccus* and *Odoribacter* from day 5 to day 40 and from day 1 to day 20, respectively. This pronounced and enduring reduction after TIM administration underscores its selective inhibitory effect on certain microbial species.

### 3.5. Correlation Analysis Between Microbiota and TIM Residue Levels in Four Tissues

Given the varied metabolism rates of TIM in different tissues, we hypothesized that its differential deposition might be linked to specific microbial interactions. Therefore, we performed correlation tests between TIM and microbes at different levels. At the phylum level, Cyanobacteria exhibited a negative correlation with the TIM residue in the kidney (Figure 7A). *Actinobacteria* and *Defferibacteres* displayed positive correlations with TIM residue in the kidney, liver, skin, and chest muscle. At the genus level, *Mucispirillum*, YRC22, Dorea, and vadinCA11 showed positive correlations with TIM residue in the kidney, liver, skin, and chest muscle (Figure 7B). Conversely, *Gallibbacterium, Parabacteroides, Odoribacter, Bilophila, Butyricimonas,* and *Faecalibacterium* displayed negative correlations with the residues in these four tissues. Notably, *Megamonas* exhibited a negative correlation solely with the TIM residue in the kidney. At the species level, *Mucispirillum_schaedleri* was positively correlated with the TIM residue in the kidney, chest muscle, skin, and liver. *Parabbacteroide_distasonis*, *Faecalibacterium_prausnitzii*, *Bacteroidess_ovatus*, *Bacteroides_fragilis*, *Bacteroides_uniformis*, *Bacteroides_eggerthii*, and *Gallibacterium_genomosp* were negatively correlated with TIM residue in the kidney, chest muscle, skin, and liver (Figure 7C).

## 4. Discussion

Previous studies suggested TIM plays a role in preventing disease in animal husbandry. As a result, a large amount of TIM was used; in 2023, TIM and its derivatives contributed to 786 million dollars in the global market. Moreover, an average of 68 mg/PCU of antimicrobials consumed by chickens contributed 33% to the global increase in antimicrobial consumption [17]. *Mycoplasma gallisepticum*, a pathogen that leads to chronic respiratory disease causing significant economic losses in the chicken industry, was effectively treated by TIM [18,19]. Previous research demonstrated TIM having greater efficacy on *Mycoplasma gallisepticum* compared to tylvalosin, doxycycline, and valnemulin according to Zhang et al. [20]. However, the use of TIM posed potential risks regarding drug resistance and food security [21,22]. In this study, we used HPLC-MS to quantify the presence of TIM residue in various tissues of Silkie chickens at different time points. Then, we examined blood biochemical parameters and found that TIM has a disruptive effect on hepatic and renal functions. Moreover, we collected cecal contents from the Silkie chickens for 16S rRNA sequencing. By correlating the presence of TIM residue with the 16S rRNA results, we identified three microbes that may contribute to the elimination of TIM.

TIM residues showed different elimination patterns in various tissues, with particularly high concentrations found in the kidney (2287ppb) and liver (2038ppb) after drug administration. These findings are consistent with those of other macrolides [23]. In contrast, the present study found the lowest concentration of TIM in the muscle (159ppb) compared to other tissues in the Silkie chicken model. Our result is consistent with previous studies on Hubbard chickens, Gushi chickens, and Bilgorajska geese [24,25,26]. Therefore, these results suggest that Silkie chickens exhibit a comparable pattern of TIM accumulation and elimination to other chicken breeds.

The accumulation of TIM also disrupted the chickens’ metabolic homeostasis, as seen in changes in blood biochemical parameters. In this study, we observed a decrease in blood urea nitrogen (BUN) and uric acid (UA). The kidneys play a crucial role in eliminating chemical waste products, including nitrogenous compounds and uric acid, from the bloodstream. An increase in blood urea nitrogen (BUN) and uric acid levels can signify a compromised renal function, where the kidneys fail to adequately filter urea and expel waste. Therefore, the observed elevation of BUN and uric acid levels following TIM treatment may suggest a potential improvement in renal function [27,28]. Moreover, hepatic function indicators such as ALP, DBIL, TBIL, and TBA showed fluctuations after TIM administration. Alkaline phosphatase (ALP), direct bilirubin (DBIL), total bilirubin (TBIL), and total bile acids (TBA) serve as critical biomarkers for assessing hepatic function, primarily secreted by the liver and typically maintained at relatively low levels in the bloodstream. In our experiment, we observed a decrease in ALP levels on days 5 and 20 in the TIM treatment group, alongside a reduction in DBIL and TBIL on day 3, followed by an increase on day 7. Notably, TBA levels increased on days 0.16 and 9. These findings do not exhibit a clear pattern of fluctuation, suggesting a complex hepatic response to TIM in the host [29,30,31].

TIM not only caused physiological changes but also led to alterations in the composition of the gut microbiome. At the phylum level, the dominant taxa identified were *Bacteroidetes*, *Firmicutes*, *Euryarchaeota*, and *Proteobacteria*. Notably, the TIM treatment group exhibited a significant increase in the relative abundance of *Bacteroidetes* on day 20, with a difference of 19.07%. Certain strains of *Bacteroidetes* are known to benefit the host by limiting pathogen colonization and enhancing immune responses [32,33]. In contrast, *Firmicutes* decreased in the TIM treatment group, with the largest difference of 12.03% observed on day 9. Many *Firmicutes* are renowned for their butyrate-producing capabilities, converting fiber into butyrate, which is crucial for pathogen defense [34,35]. The reduction in *Firmicutes* in the TIM treatment group may consequently lead to decreased butyrate levels, warranting further investigation. *Euryarchaeota*, which decreased by 10.62% in the TIM treatment group, are generally considered commensal or beneficial organisms, performing immunomodulatory functions by promoting cytokine production [36]. Further exploration is required to elucidate the effects of *Euryarchaeota* on Silkie chickens. Additionally, *Proteobacteria* increased by 3.31% on day 20 and 2.68% on day 120; however, given that *Proteobacteria* are recognized as biomarkers of disease in humans [37], this increase may have adverse implications for the health of the TIM treatment group.

On the genus level, our study observed a decrease in the relative abundance of *Bacteroides* five days after TIM administration. Previous research has emphasized the important role of *Bacteroides* in supplying nutrients and short-chain fatty acids to other microbes, as well as protecting the host from pathogens and inflammation [38,39]. The reduction in *Bacteroides* abundance suggested that TIM treatment may decrease pathogen protection. Additionally, the relative abundance of *Methanobrevibacter* decreased three days after TIM treatment. Research by Manhert and Bernachez shows a positive correlation between *Methanobrevibacter* and inflammation [40,41]. The decrease in *Methanobrevibacter* after TIM treatment may inhibit inflammation. Another microbe that exhibited significant changes was *Oscillospira*, which has been strongly linked to diabetes, obesity, and human health [40,42]. The reduction in *Oscillospira* after TIM treatment may contribute to an increased risk of obesity-related metabolic diseases. 

Due to the widespread use of antibiotics in poultry, which can accumulate in food products [43,44], there is a resultant concern regarding strain mutations and the increased antibiotic resistance of pathogens [45]. Accordingly, we conducted an analysis to examine the correlation between the gut microbiome and TIM residues, with the aim of identifying potential strains that may mitigate the accumulation of TIM in the environment. On the species level, *Parabbacteroide_distasonis* and *Bacteroides uniformis* were negatively correlated with TIM residue in all tissues. *Parabacteroide distasonis* was found to suppress non-alcoholic steatohepatitis and hepatocyte pyroptosis [46,47]. Combining these results, the negative correlation with TIM residue in the liver suggested an important role of *Parabacteroide distasonis* in modulating hepatic function. *Bacteroides uniformis* had an anti-inflammatory function in the gut, according to Dai and Yan [48,49]. Our finding of *Bacteroides uniformis* correlation with antibiotic elimination may provide new insight into *Bacteroides uniformis* metabolic function. In summary, TIM treatment can induce changes in both pathogenic and probiotic microorganisms. Therefore, further research is needed to explore the role of gut microbes in the elimination of TIM residue.

## 5. Conclusions

In this study, we discovered different elimination patterns among tissues in Silkie chickens and identified a strong correlation between TIM residue and gut microbes. We observed a significant positive correlation between *Mucispirillum_schaedleri* and TIM residue across all tissues, whereas *Parabbacteroide_distasonis*, *Faecalibacterium_prausnitzii*, and other microbes displayed negative correlations with TIM residues. Furthermore, our data confirmed that using TIM will have a long-term impact on gut microbial balance in chickens. These findings will offer valuable insights into the application of TIM in poultry production.

## Figures and Tables

**Figure 1 animals-14-03428-f001:**
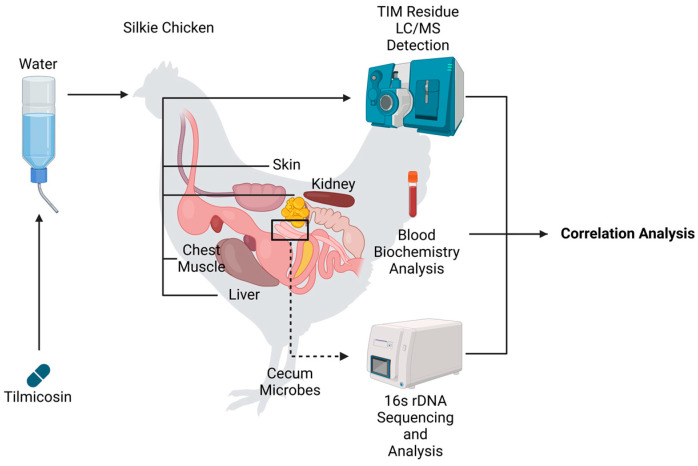
Silkie chickens were subjected to TIM or control treatment. Blood, left thorax skin, left chest muscle, liver, and cecum feces were collected for further experiments and correlation analysis.

**Figure 2 animals-14-03428-f002:**
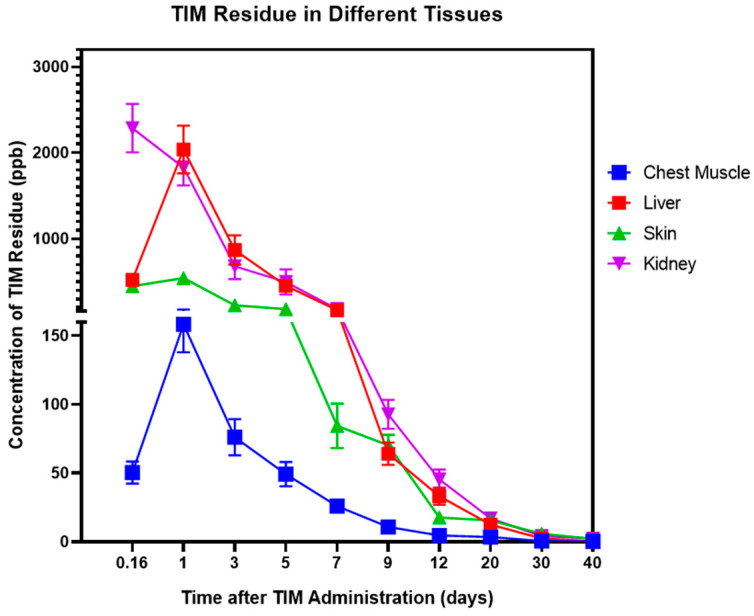
The pharmacokinetics of TIM in Silkie chickens. The concentrations of TIM in the chest muscle, skin, liver and kidney. A sample size of *n* ≥ 8 was used for each group, with data presented as mean ± SEM.

**Figure 3 animals-14-03428-f003:**
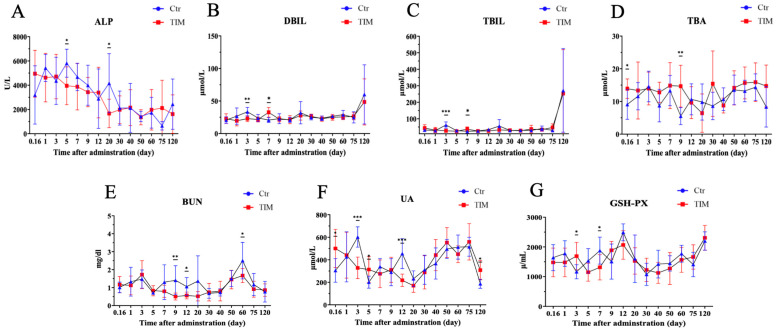
Biochemistry parameters in Silkie chickens post TIM administration. (**A**) Change in serum alkaline phosphatase (ALP). (**B**) Direct bilirubin (DBIL). (**C**) Total bilirubin (TBIL). (**D**) Total bile acid (TBA). (**E**) Blood urea nitrogen (BUN). (**F**) Uric acid (UA). (**G**) Glutathione peroxidase (GSH-Px). Statistical significance is indicated as * *p* < 0.05, ** *p* < 0.01, *** *p* < 0.001; determined by unpaired *t*-test. “Ctr” stands for “controls” (*n* ≥ 5); “TIM” stands for “TIM-treated Silkie chickens” (*n* ≥ 8).

**Figure 4 animals-14-03428-f004:**
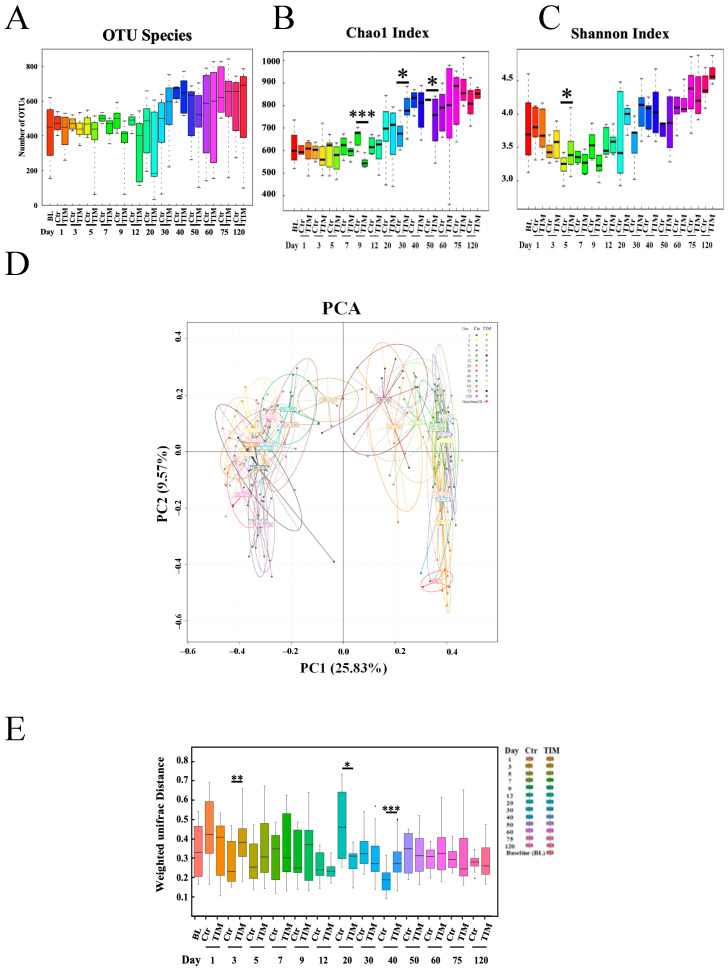
Changes in the gut microbiota of Silkie chickens at the OTU level. (**A**) OTU counts of the TIM-treated (TIM) and control (Ctr) groups at various time points. (**B**) Bacterial richness differences between the control and TIM-treated Silkie chickens at various time points were assessed using the Chao1 index. (**C**) α-diversity in the Ctr and TIM groups was evaluated using the Shannon index. (**D**) β-diversity comparisons were made via principal component analysis between the Ctr and TIM groups over time. (**E**) The weighted Unifrac distance was calculated between the two groups, Ctr and TIM. D, day; Ctr, controls (*n* ≥ 5); Statistical significance is indicated as * *p* < 0.05, ** *p* < 0.01, *** *p* < 0.001; determined by unpaired *t*-test. TIM, TIM-treated Silkie chickens (*n* ≥ 8).

**Figure 5 animals-14-03428-f005:**
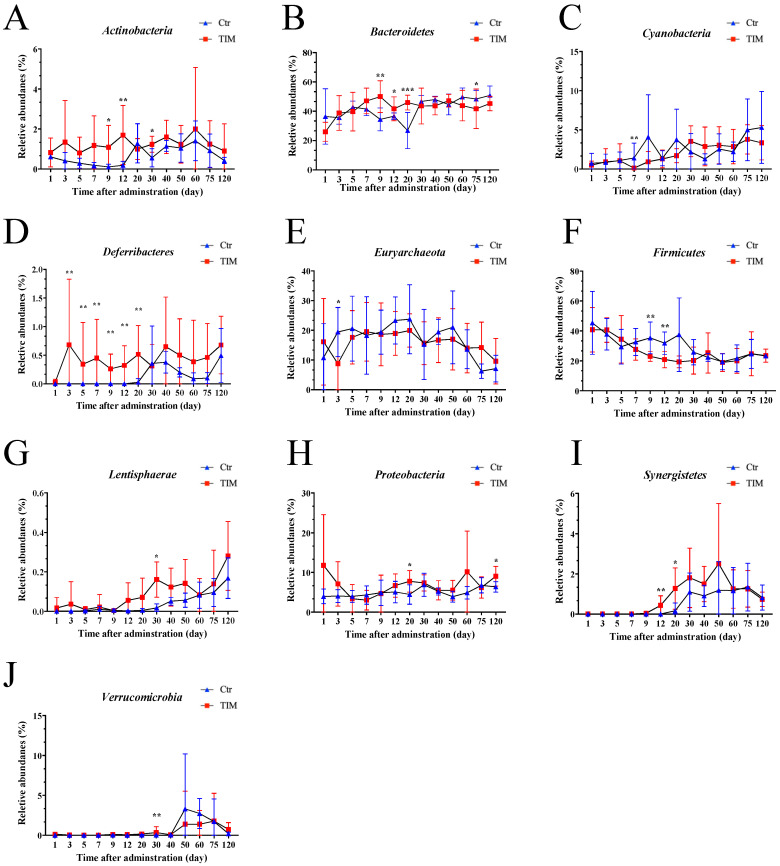
Relative abundances of the significantly changed gut microbiota at the phylum level after the withdrawal period. (**A**) *Actinobacteria* in the control and the TIM-treated group. (**B**) *Bacteroidetes* in the control and TIM-treated groups. (**C**) *Cyanobacteria* in the control and TIM-treated groups. (**D**) *Deferibacteres* in the control and TIM-treated groups. (**E**) *Euryarchaeota* in the control and TIM-treated groups (**F**) *Firmicutes* in the control and TIM-treated groups. (**G**) *Lentisphaerae* in the control and TIM-treated groups. (**H**) *Proteobacteria* in the control and TIM-treated groups. (**I**) *Synergistes* in the control and TIM-treated groups. (**J**) *Verrucomicrobia* in the control and TIM-treated groups. Statistical significance is indicated as * *p* < 0.05, ** *p* < 0.01, *** *p* < 0.001, assessed using the Wilcoxon rank-sum test. Ctr, controls (*n* ≥ 4); TIM, TIM-treated Silkie chickens (*n* ≥ 6).

**Figure 6 animals-14-03428-f006:**
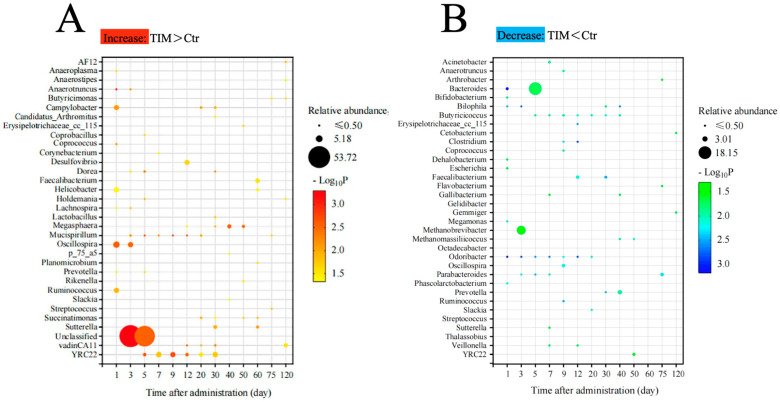
Changes in the gut microbiota relative abundances were observed in response to TIM treatment. (**A**) Gut microbiota that increased in relative abundance in the TIM-treated group. (**B**) Gut microbiota with decreased relative abundance in the TIM-treated group. Analyzed using the Wilcoxon rank-sum test. Ctr, controls (*n* ≥ 4); TIM, TIM-treated Silkie chickens (*n* ≥ 6).

**Figure 7 animals-14-03428-f007:**
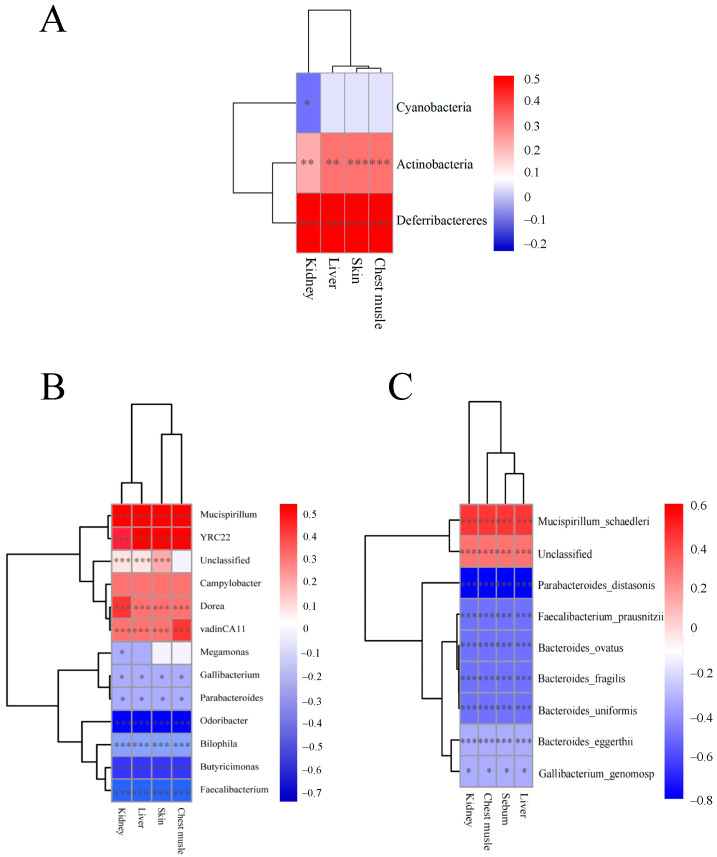
Significant correlations were found between gut microbiota and residual TIM in various tissues. (**A**) Pearson’s correlation at phylum level. (**B**) Pearson’s correlation at genus level. (**C**) Pearson’s correlation at specie level. Statistical significance indicated as * *p* < 0.05, ** *p* < 0.01, *** *p* < 0.001, assessed using Pearson’s test. Pearson’s correlation coefficients (R) displayed as side values with corresponding *p*-value * for indicating Pearson’s linear correlation (P). Ctr, controls (*n* ≥ 4); TIM, TIM-treated Silkie chickens (*n* ≥ 6).

## Data Availability

Raw sequence data are available under the National Center for Biotechnology Information (NCBI) SRA accession number PRJNA817361.

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
