# Peer review of "Dynamic Shifts in Antibiotic Residues and Gut Microbiome Following Tilmicosin Administration to Silkie Chickens"

_animals, 2024, doi:10.3390/ani14233428_

Round 1

Reviewer 1 Report

Comments and Suggestions for Authors

The manuscript entitled " Dynamic Shifts in Antibiotic Residues and Gut Microbiome Following Tilmicosin Administration Throughout Silkie Chicken Development" by Qiying Liang et al. The researcher addresses a topic of interest to Animal Journal readers and the poultry industry. The work tested the important role of gut microbiota in influencing antibiotic metabolism, providing a potential strategy for reducing drug residues in animal food and ensuring food safety. Although I find this manuscript commendable, some items need revision /clarity.

Major

1. The author should give the diet composition fed to the chickens during the experimental and post-experimental periods (different time points) for a better understanding of the work and repeatability/reproducibility of the work.

2. The author needs to justify why 0.75g/L tilmicosin is added to drinking water. Why not a lower or higher level? Maybe use past articles to support this.

3. The discussion requires a more thorough review of relevant research with a stronger focus on linking the different results for a better understanding of TIM residues and their effects on chickens. Several arguments presented lack sufficient depth to support this study's results adequately.

Specific comments

L81, is there any reason for using “female” silky chicken?

L85, what is the meaning of “CA”?

L87, can you correct “10ten”

L439, can you correct “6six”

Author Response

Dear Reviewer, thank you for your comments. Please find our responses in the attached document.

Reviewer 2 Report

Comments and Suggestions for Authors

Dear Editor,

Antibiotic usage for treatment and growth promotion has become integral part of intensive livestock and poultry rearing. Though Antibiotic free production has been implemented many countries, responsible use of antibiotics in food animal production is the emerging trend in order to alleviate the suffering of animals and also ethical part of animal production. Poultry industry is increasingly relying on macrolide antibiotics such as Tylosin, Tilmicosin, and Tilvalosin for mycoplasma control. However less known about interaction of these antibiotics with intestinal microflora which plays a huge role in many physiological functions of the body. In this context the manuscript is well written however the following suggestion need to be carried out before final acceptance of the paper.

Line no- 21 - TIM treatment –Full form.  

Abstract:

Line nos-34-35- Decreased serum levels of TBA, BUN, and UA, while increasing the levels of DBIL, TBIL, and GSH-PX at day 3, followed by a decrease from day 5 onwards.  – The abbreviations should be written in full form

Introduction:

Line no 50-51: Its approval for treating respiratory diseases in chickens caused by Mycoplasma gallisepticum and Mycoplasma hyopneumoniae highlights its practical significance- Mycoplasma hyopneumoniae is the pathogen of pigs so it should be removed, may be Mycoplasma synoviae could be added

Materials and methods:

Line no-85- CA concentration of 0.75g/L tilmicosin was ad- 85-What do you mean by CA concenteration

Line no-87- , 10ten chickens – Correct and give in either in number or in word

Line no-87-  6six chickens  - Correct and give in either in number or in word

It would be better if the authors justify why they have chosen Silky chicken over meat type /egg type chickens

Discussion part:

The discussion part is very brief. The authors should add information about global usage of antibiotics in poultry with special reference to tilmicosin. Further discussion is needed how the accumulation of antibiotics in chicken tissues will negatively impact animal/public and environmental health. That is more important.

The points discussed in line nos 335-340 need to be elaborated in detail and the ill effects compared with other antibiotics whether tilmicosin have similar effects or more or less

Regarding microflora the authors should add discussion regarding importance all four beneficial genera Bacteroidetes, Firmicutes, Euryarchaeota, and Proteobacteria  and their correct ratios, as well as their beneficial effects also.

Author Response

(The authors gave the same response as above.)

Round 2

Reviewer 2 Report

Comments and Suggestions for Authors

The authors have carried out necessary corrections and significantly improved the manuscript  as suggested. Hence accepted. 

Author Response

Thank you for your review. We are truly appreciate for your comments and suggestion.